# An Explanation about the Use of (*S*)-Citronellal as a Chiral Derivatizing Agent (CDA) in ^1^H and ^13^C NMR for *Sec*-Butylamine, Methylbenzylamine, and Amphetamine: A Theoretical-Experimental Study

**DOI:** 10.3390/molecules24152830

**Published:** 2019-08-03

**Authors:** Viviani Nardini, Vinicius Palaretti, Luis Gustavo Dias, Gil Valdo José da Silva

**Affiliations:** Departamento de Química, Faculdade de Filosofia, Ciências e Letras de Ribeirão Preto, Universidade de São Paulo, Avenida dos Bandeirantes, 3900, Ribeirão Preto 14040-901, SP, Brazil

**Keywords:** long-chain imine, conformation, NMR, NOE

## Abstract

A chiral derivatizing agent (CDA) with the aldehyde function has been widely used in discriminating chiral amines because of the easy formation of imines under mild conditions. There is a preference for the use of cyclic aldehydes as a CDA since their lower conformational flexibility favors the differentiation of the diastereoisomeric derivatives. In this study, the imines obtained from the reaction between (*S*)-citronellal and the chiral amines (*sec*-butylamine, methylbenzylamine, and amphetamine) were analyzed by the nuclear Overhauser effect (NOE). Through NOE, it was possible to observe that the ends of the molecules were close, suggesting a quasi-folded conformation. This conformation was confirmed by theoretical calculations that indicated the London forces and the molecular orbitals as main justifications for this conformation. This conformational locking explains the good separation of ^13^C NMR signals between the diastereomeric imines obtained and, consequently, a good determination of the enantiomeric excess using the open chain (*S*)-citronellal as a CDA.

## 1. Introduction

The use of nuclear magnetic resonance (NMR) to determinate the purity of an enantiomer is based on a principle in which a pair of enantiomers of the studied substrate is converted into a pair of diastereoisomers through of the addition of chiral derivatizing agents (CDAs) or chiral solvating agents (CSAs) (Figure 1).

If a reaction between the CDA and the mixture of enantiomers occurs with 100% conversion, it is possible to relate the ratio of diastereomers observed in the NMR spectrum to the ratio of enantiomers.

Chiral agents used for the derivatization of chiral amines which form diastereomeric amides, such as Mosher’s reagent (α-methoxy-α-trifluoromethylphenylacetic acid (MTPA)) and α-methoxyphenylacetic acid (MPA), are effective in resolving the signals of the derivatives in the NMR spectrum [1,2].

Among these CDAs, it is possible to add, inside the NMR tube, some alkaline metal salt—for example, barium perchlorate (Ba(ClO_4_)_2_)—in order to cause the cation (Ba^2+^) to form a complex with the oxygen atoms of the methoxy and carbonyl groups. This addition favors one of the conformations. This conformational constraint is also expected when the temperature is decreased [3].

In other words, the procedure of adding chelating metals is one way of predicting the preferential conformation of the diastereomeric derivatives formed with the chiral derivatizing agents.

In the case of (*R*)-MPA, the evidence shown in the ^1^H NMR spectra together with theoretical calculations suggest that the Ba^2+^ binds strongly to the oxygen atoms of the carbonyl and methoxy groups, leading to an increase in the stability and population of the sinperiplanar conformer [3].

In these cases, a reduction in the number of conformers provides a more resolved signal for each diastereomeric derivative, which results in more resolved signals (popularly called finer signals). This event gives rise to a greater chemical shift difference between the two diastereoisomers (Δ*δ*) (Figure 2). Consequently, the integral that helps to quantify the isomers can be more accurately measured, as exemplified in the figure below.

Since it is known that the NMR signal of a given molecule is the weighted average of the chemical shifts of all relevant conformers, according to the Boltzmann distribution [4], researchers who develop CDAs usually tend to restrict the molecular motion by including cyclic compounds or hydrogen bonds to decrease the number of conformers.

In addition to the use of the metals to favor and even predict the preferential conformation of the diastereomeric derivatives, other chiral derivatizing agents can also be exploited, making use of the formation of intramolecular hydrogen bonds to force a preferential conformation.

This approach can be applied in CDAs such as (*R*)-*N*-(2-nitrophenyl) proline, in which an intramolecular hydrogen bond between the NH of the imine and the oxygen of the NO_2_ is observed [5]. Another example occurs in (1-naphthyl)(trifluoromethyl) *O*-carboxy anhydride, which reacts with primary amines, giving α-hydroxyamides that tend to remain in an eclipsed conformation due to the intramolecular hydrogen bond [6].

It is also possible to include the aldehyde function in chiral derivatizing agents which, in the presence of chiral amines, give imines under mild conditions. Generally, these imines can be analyzed by NMR without purification and/or additional reagent, as occurs in the formation of amides [7,8].

Our research group has obtained imines as the reaction products from the condensation of the terpene (*S*)-citronellal with chiral primary amines. In our most recent study [9], we used the imines to quantify the enantiomeric excess and to establish the absolute configuration of aliphatic and aromatic amines by ^1^H and ^13^C NMR. This analytical method proved to be fast and useful to quantify *sec*-butylamine, methylbenzylamine, and amphetamine upon reaction with (*S*)-citronellal inside the NMR tube. Moreover, it rapidly produced the corresponding diastereomeric imines (Scheme 1).

As far as we are aware, there are no reports in the literature of CDAs making use of acyclic compounds due to the lack of conformational locking to improve the chemical shift difference between the diastereoisomers (Δ*δ*). This led us to the following question: how can the efficacy of (*S*)-citronellal as a CDA of chiral primary amines be explained?

Therefore, this study investigates the conformations of imines 5, 6, and 7 (Scheme 1) by NMR and theoretical calculations in an attempt to explain why (*S*)-citronellal is a good CDA for the NMR assignment of the absolute configuration of primary amines.

## 2. Results and Discussion

In a previous paper [10], we discussed the conformation of citronellal, the precursor of the imines shown in Scheme 1. In that paper, theoretical calculations and NOE experiments revealed that citronellal preferentially assumes a folded conformation.

NOE provides relevant information about molecular motions and intra- and intermolecular distances between atoms. Changes in the intensity of the NOE signal enhancement result from the transfer of polarization between dipolarly coupled nuclei via spin-lattice relaxation mechanisms (T1) [11]. In the case of citronellal [10], the hydrogens at both ends of the molecule present an NOE, which is consistent with a folded conformation, as verified by the theoretical calculations (Figure 3).

As described in the introduction of this article, we previously used (*S*)-citronellal as an effective CDA to quantify the enantiomeric excess of chiral primary amines [9]. Our studies indicated that the imines arising from the reaction between (*S*)-citronellal and racemic amines display well-resolved signals in both their ^1^H and ^13^C spectra, which allows us to distinguish between *S,S* and *S,R* diastereoisomers clearly [9]. Therefore, we became interested in studying the conformations of these diastereomeric imines to understand their behavior.

We started to investigate the conformation of imines by carrying out NOE experiments with an equimolar mixture of the *S,S* and *S,R* diastereoisomers of compounds **5**, **6**, and **7**. Since NOE does not have an instantaneous effect on the atoms and requires some time to develop, several studies have been conducted with varying the development time, called the mixing time, of NOE to determine the appropriate experimental conditions.

In compound **5**, H11 irradiation at two different mixing times afforded NOE between H11-H1 and H11-H8 (Figure 4).

This same vinyl hydrogen (H11) is saturated in compounds **6** and **7**, so NOE with aromatic hydrogens arises even though these hydrogens are far from H11 (Figure 5 and Figure 6). H11 irradiation also provides NOE between H11 and H8 in both compounds, which is confirmed when H8 is irradiated. These results indicate that the preferred conformation of these long-chain imines is quasi-folded, different from citronellal which is completely folded.

To find a theoretical-experimental correlation, we performed a conformational search and obtained a large number of conformers for each of the diastereoisomers (Appendix A).

Inspection of the structures of the most stable conformers (Appendix A) shows that quasi-folded conformations predominate, corroborating the result observed in the NOE experiments.

NOE enhancement occurs when the distance between hydrogens does not exceed 4.0–5.0 Å [12]. Therefore, we measured the distances between hydrogen atoms in all conformers of each imine to interpret the NOE experiment. As observed in Appendix A, values lower than 5.0 Å for the conformers corroborated with the NOE experimental data.

In our study of citronellal [10], the driving force behind the preference for the folded conformation was the intramolecular dispersion interaction, which we obtained during the theoretical calculations using the D3 correction developed by Grimme [13,14,15].

In the case of imines, we calculated the energy of each conformer without D3-correction to verify the importance of London interaction in ranking the conformers. We realized that most of the more stable conformers are not folded when we did not apply the D3-correction (Figure 7), a situation that is at odds with the experimental data.

The imines under study bear two reaction sites where chemical reactions can occur: the double bond, which refers to the imine function, and the carbon–carbon double bond. These sites may be viewed as Lewis acid and Lewis base, respectively.

Another approach, established by Fukui and co-workers, proposes that certain reactions can be predicted by frontier molecular orbitals (FMOs) [16,17,18]. Similarly, we would be comparing the Lewis base and the Lewis acid with the highest occupied molecular orbital (HOMO) and the lowest unoccupied molecular orbital (LUMO), respectively. These orbitals are very useful in chemistry because the energy gap between the HOMO and LUMO describes the chemical reactivity, optical polarizability, kinetic stability, and chemical softness/hardness of an acid or base. However, according to Fukui, the HOMO or LUMO may not be suitable for a given reaction, but the next orbital, or any orbital that has energy very close to that of the HOMO or LUMO, could be used [16,17,18].

Taking this into consideration, the frontier effective-for-reaction molecular orbital (FERMO) concept was introduced to identify which molecular orbital is actually involved in the reaction. In other words, the FERMO was constructed to solve the constraints of the HOMO and LUMO. To this end, the composition of the molecular orbital, its format, and a little critical chemical sense are taken into account [19,20]. The FERMO concept is being widely used to describe acid–base behavior in organic and inorganic complexes, pericyclic reactions, and biological systems [21].

On the basis of this new system that focuses not only on the HOMO and LUMO, we analyzed 12 orbitals (energy diagrams of the molecular orbitals are shown in Appendix A) to find an additional justification, besides D3-correction, for the quasi-folding of the imine molecule. In other words, we focused on “reaction sites” (C=N and C=C) because the acid–base interaction between these sites is related to the HOMO (electron donor) and the LUMO (electron acceptor).

The orbitals present in the carbon–carbon double bond and in the imine function of imines interact. These orbitals are common to the six compounds analyzed here (5*_S,S_*, 5*_S,R_*, 6*_S,S_*, 6*_S,R_*, 7*_S,S_*, and 7*_S,R_*). The orbitals localized in these “reaction sites” are important because they govern the molecule acid–base character and cyclization, for example.

After extensive investigation, we selected the interactions with the lowest energy gap between the molecular orbitals (Figure 8).

For the overlap of the molecular orbitals shown in Figure 8, we considered the smaller energy gap and the phase similarity of the orbitals. In the case of the investigated imines, overlap of the molecular orbitals stabilizes the quasi-folded conformation when the energy gap and phase are adequate. Therefore, we can conclude that the phase interaction between the molecular orbitals favors the quasi-folding conformation of the imines.

Therefore, we suggest that there is a certain conformational locking even in the absence of cyclic compounds or hydrogen bonds in the analyzed imines.

To observe this effect of conformational locking obtained by the interaction of molecular orbitals and London forces, we superimposed the first three most stable conformers of the imine derived from *sec*-butylamine (Figure 9). Thereby, it was possible to verify that these conformers have similar conformation, giving rise to distinct NMR signals from *S,S* and *S,R* diastereoisomers.

Therefore, we conclude that the NMR signals of diastereomeric derivatives (*S,S* and *S,R* imines) are well separated because the conformers do not differ significantly due to the quasi-folded conformation caused by London forces and interactions between molecular orbitals.

## 3. Materials and Methods

In each experiment described in our previous work [9], 20 μL of (*S*)-citronellal (96%, 0.11 mmol, (Sigma-Aldrich, Tokyo, Japan) was placed in an NMR tube (5 mm outer diameter) and 1.1 to 1.3 equivalents of the chiral racemic amine were added (*sec*-butylamine, methylbenzylamine, or amphetamine). Deuterated solvent (0.5 mL of CDCl_3_) was only added a few minutes later (2–3 min) to prevent the solvent from affecting the reaction.

The 1D ^1^H NMR and ^13^C{^1^H} NMR spectra were acquired at a temperature of 298 K on a Bruker DRX 500 instrument (B_o_ = 11.7 T, Fällanden, Switzerland).

The chemical shifts are given relative to the internal TMS or to the residual ^1^H signal of the solvent (CDCl_3_). The ^1^H NMR spectra were acquired by using 16 transients into 64 k data points (pulse sequence *zg30*). Fourier transform was applied to the FID (free induction decay) without the use of any window function (line broadening = 0 Hz). Other parameters were: SI (size of real spectrum) = 32 k, DS (number of dummy scans) = 2, SW (spectral width) = 12.0 ppm, RG (receiver gain) = 25.4, and AQ (acquisition time) = 5.34.

The 1D broadband ^1^H-decoupled ^13^C NMR spectra were obtained by using 1.5 k transients into 64 k data points, which were processed with exponential multiplication (line broadening = 1.0 Hz) of the FID (pulse sequence *zgig30*). Inverted-gated decoupling was used during ^13^C{^1^H} NMR; the decoupler was turned off during the recovery delay. Other parameters were: SI (size of real spectrum) = 32 k, DS (number of dummy scans) = 4, SW (spectral width) = 250.0 ppm, RG (receiver gain) = 32 k, and AQ (acquisition time) = 1.04.

^1^H and ^13^C NMR spectra and spectroscopic data of the imines are presented in the Appendix A.

1D NOESY spectra were acquired at 298 K on a Bruker DRX 500 instrument (B_o_ = 11.7 T) in CDCl_3_ solvent (pulse sequence *selnogp*). The number of scans (NS) was 256. Four mixing times (0.3, 0.4, 0.5, and 0.6 s) were tested to determine the optimum nuclear Overhauser effect (NOE) accumulation value. Low-power RF pulses (60 dB attenuation) can slowly saturate protons. Other parameters were: SI (size of real spectrum) = 64 k, DS (number of dummy scans) = 4, SW (spectral width) = 17.0 ppm, RG (receiver gain) = 18.0, AQ (acquisition time) = 1.92, LB (line broadening) = 0.3, and GB (Gaussian broadening) = 0.3.

The conformational search (GMMX) was performed with PCModel version 7.0 [22]; the parameters of our previous work were used. Because imines preferentially form the *E*-isomer due to lower steric hindrance, the torsional angle defined by the C–C=N–C atoms of compounds **5**, **6**, and **7** was restricted to limit the conformational search to this isomer [23,24,25].

After the conformational search, each of the most stable conformers was re-optimized with ORCA version 3.0.1 [26] at the B3LYP-D3(BJ)/def2-TZVP(-f) level [27,28,29,30,31,32]. Dispersion interaction between the atoms was taken into account by following the approach of Grimme and co-workers. Tight criteria for the SCF cycle and geometry were employed [13,14,15].

The conformers were ranked according to the Boltzmann distribution [33] (Equation (1)) and the structures were visualized using GaussView version 4.1.
(1)$$\%\;{\mathrm{conformer}}_{i}\;=\;100e^{\left(-\Delta Ei/RT\right)}/\sum_{j=1}^{n}e^{\left(-\Delta Ej/RT\right)}.$$

## 4. Conclusions

In 1981, the cyclization of imines derived from the reaction between (+)-(*R*)-citronellal and phenylmethanamine in the presence of tin(IV) chloride-catalyst was studied by Demailly and Solladie [34]. These authors suggested that imine presented a folded conformation, which was termed quasi-chair conformation due to the products arising from the reaction. However, this assumption had yet to be confirmed by theoretical chemistry or computational calculations until the publication of our work.

Almost 40 years later, as far as we know, we are the first to conclude that the structures of the target imines exhibit quasi-folded conformations in vacuum and in solution after being mapped by nuclear magnetic resonance (NMR) spectroscopy and computational calculations.

In addition to defining the conformations of these imines, we were able to find an explanation for this through the dispersion forces and interactions between molecular orbitals that contribute to the preferential quasi-folded conformation.

With these results obtained, we were able to show the reason why (*S*)-citronellal is a good chiral derivatizing agent for primary chiral amines—once the conformers of the diastereomeric imines are made highly similar due to the conformational locking derivative cause by the effects of dispersion forces and interactions between molecular orbitals, there is a reduction in the number of conformers, which contributes to a more resolved signal for each diastereomeric derivative. Consequently, the quantification of the isomers through the integral is more accurately measured.

With the statements obtained in our studies, it will be possible to computationally “draw” other chiral derivatizing agents (CDAs) that have a restricted number of conformers and optimize the experimental part for better orientation in the choice of CDAs.

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
