# Peer review of "An Explanation about the Use of (S)-Citronellal as a Chiral Derivatizing Agent (CDA) in 1H and 13C NMR for Sec-Butylamine, Methylbenzylamine, and Amphetamine: A Theoretical-Experimental Study"

_molecules, 2019, doi:10.3390/molecules24152830_

Round 1

Reviewer 1 Report

This paper is the continuation of a previous paper from the same authors. They describe an interesting use of a chiral aldehyde to resolve chiral amines by converting them into diasteroisomeric imines. The authors studied the conformation of such imines using NOE-nmr and theoretical calculations. This work may be useful to organic and inorganic chemistry researchers dealing with chiral compounds.

Here is my comment:

Strenght: this study may be helpful in the chiral compounds identification strategies. In fact, it seems this method allows discrimination of several racemic amines transforming them in diastereoisomers.

Weakness: there are some required improvements: the nmr data must be reported more clearly (all signals are reported as "Overlapping signals" but this is not true). The molecule number must be reported in the Supp Info (this part is somehow difficult to follow).

You could find the revised files with some correction and comment in attached file.

Author Response

We acknowledge and accept the comments and corrections described in the article by reviewer 1. All corrections have been made.

In this paper the spectra obtained in the reaction between the (S)-citronellal and the racemic mixtures of the amine are present.

It is important to note that detailed elucidation of the multiplicity of each 1H NMR signal was not performed for the following reasons:

1) Because they are spectra with two diastereoisomers and not only one.

2) In addition, in each diastereoisomer formed there is a large number of diastereotopic hydrogens that also make the analysis difficult.

3) The signs seem to be well resolved, but in fact it is an overlapping of signals. For example, the signal at 5.05 ppm (H11 of compound 5) in Figure S1 appears to be a triplet, but is (at least) two doublet of doublets (since H10a and H10b are diastereotopics hydrogens), as exemplified below.

Figure 1. Explanation about overlapping signal of H11 from imines 5S,R and 5S,S.

4) We obtained a Jresolved experiment (Jres) to see the overlapping signals as showed in the signal at 5.05 ppm (H11 of compound 5):

                                             Figure 2. Jresolved of H11 from imines 5S,R and 5S,S.

As noted in this example above, the signals of these molecules are not simple and correspond, in fact, to an overlap of signals and couplings.

5) Finally, there was no need to identify the exact multiplicity of the signals, since the signals of interest (H11, H1, H8 in all compound 5, 6 and 7) are an easily identifiable in the spectrum.

Regarding the corrections suggested by reviewer 1 in Supplementary Information, we identify the molecules in the figures, tables and spectra.

In table S1, the H1 mu1plicity is actually two triplets, as proposed by reviewer 1. In the same table S1, H13 and H14 are not singlets, since, as mentioned, there are two diastereoisomers, i.e. two H13 (H13S,S and H13S,R) and two H14 (H14S,S and H14S,R).

In table S2, the H1 mu1plicity is actually two doublets.

Reviewer 2 Report

In this paper, the authors explore the use of (S)-citronellal as Chiral derivatizating agent to obtain  imines starting  from chiral amines (sec-butylamine, methylbenzylamine and amphetamine). They analysed the derived diastereoisomers  by nuclear Overhauser effect (NOE) NMR experiments and tried to rationalise NMR data by theoretical calculations.

The work is interesting and the approach is also convincing but the obtained data are not clearly presented and the conclusions are vague.

The NOE interactions (as derived from 1D experiments) are described only in qualitative and not in quantitative terms. Have the authors tried to perform also 2D NOESY experiments in order to quantify the NOE effects?

In an equilibrium situation, the presence of a NOE with low intensity can be related to a conformer with a low population. 

The NOE correlations are poorly related with the conformer identified by theoretical approach, and it means the conclusion difficult to support (i.e. in Figure 9, the protons are omitted, making it not useful to understand if that conformers are in agreement with NMR data). A quantitative correlation of NOE intensities to the presence of different conformers is necessary.

The figures are not correctly cited into the text making difficult to follow the flow of information.

The comment about NOEs of compound 6 and 7 is the same, but the presence of NOE H11/H8 are true only for compound 7 (in compound 6, it seems an artifact).

The figure legend did not explain correctly the figure content (and sometimes the captions contain English errors).

Materials and Methods poorly described the NMR methods.

In Supplementary information, also the number of the molecules should be added in  the figures.

Author Response

At the suggestion of the reviewer, we performed 2D-NOESY experiment using the pulse sequence noesyph with the parameters TD = 2048 (F2) and 256 (F1), DS = 4 and NS = 24 (Figures 1, 2 and 3). However, 2D-NOESY presented some artifacts and noise that could make signal attribution inaccurate, which seems 1D-NOESY is a better choice, providing cleaner spectra with an easy and direct interpretation. As mentioned on the article of Butts and co-workers (Org. Biomol. Chem., 2011, 9, 177): “in summary, we find the 1D-NOESY experiment to be faster, simpler, at least as accurate as 2D-NOESY for internuclear distance determinations, and able to applied with more confidence to longer range NOE contacts.”

We know that the usual choice is 2D-NOESY with zero-quantum suppression, but exceptions would be if you have a very concentrated sample and you are only interested in one or two NOEs and the peaks to be irradiated are well-separated, as is our case.

The purpose of running a NOESY is often to determine conformation by establishing a distance between two protons. When more than one conformation is present, however, the NOE may give misleading distances. If the conformations are being averaged over the time scale of the experiment (the mixing time) the NOE will not reflect the average distance between the protons but rather the average of the inverse sixth power of the distance.

Figure 1. 2D NOESY (pulse sequence noesyph, T = 300 K) from imines 5S,R and 5S,S.

Figure 2. 2D NOESY (pulse sequence noesyph, T = 300 K) from imines 6S,R and 6S,S.

Figure 3. 2D NOESY (pulse sequence noesyph, T = 300 K) from imines 7S,R and 7S,S.

In the attempt to verify the NOE through the 2D-NOESY experiment, we performed this same experiment at low temperature (273 K), as lower temperatures should favor the observation of the lowest energy conformers. However, no additional results were found, as can be seen in the following figures.

Figure 4. 2D NOESY (pulse sequence noesyph, T = 273 K) from imines 5S,R and 5S,S.

Figure 5. 2D NOESY (pulse sequence noesyph, T = 273 K) from imines 6S,R and 6S,S.

Figure 6. 2D NOESY (pulse sequence noesyph, T = 273 K) from imines 7S,R and 7S,S.

Thus, we performed quantitative NOE measurements through 1D-NOESY based on the relationship where the NOE intensity is inversely proportional to the sixth power of the distance.

Figure 7. 1D NOESY (pulse sequence selnogp) from imines 5S,R and 5S,S.

Figure 8. 1D NOESY (pulse sequence selnogp) from imines 6S,R and 6S,S.

Figure 9. 1D NOESY (pulse sequence selnogp) from imines 7S,R and 7S,S.

Table 1. Quantitative NOE of the imines 5, 6 and 7

Compounds

Integral (see Figures 7, 8 and 9)

Distance found through integral

5S,R and 5S,S

H8 = 0.0010

3.21

5S,R and 5S,S

H1 = 0.0010

3.16

6S,R and 6S,S

H8 = 0.0016

3.42

7S,R and 7S,S

H8 = 0.0023

3.63

In the Materials and Methods, we have added more information on spectral acquisition such as pulse sequence, size of real spectrum, number of dummy scans, spectral width, receiver gain, acquisition time, line broadening and Gaussian broadening. We hope that there will be an enrichment of this section describing the NMR data.

In Supplementary Material, compound structures and numbering were added for better identification.

In Figure 9 we do not place the hydrogens to make the visualization of the molecule easier. With the addition of the hydrogens, we realized that it was not possible to verify clearly that the more stable conformers presented almost the same conformation. This can be confirmed in the phrase already mentioned in the paper: “To observe this effect of conformational locking obtained by interaction of molecular orbitals and London forces, we superimpose the first three most stable conformer of the imine derived from sec-butylamine (Figure 9).”

In compound 6 we can confirm that the NOE found between H11 (5.07 ppm) and H8 (0.91 ppm) is not an artifact, since we performed the irradiation in H11 and also in H8. When we performed the irradiation in H11 (5.07 ppm), we observed the NOE in H8 (0.91 ppm) and when we performed the “inverted NOE”, irradiating in H8 (0.91 ppm), we observed NOE in H11 (5.07 ppm). That is, there is no doubt that NOE really exists.

Reviewer 3 Report

Authors have described the study on diastereomeric mixture of imines obtained from the reaction between (S)-citronellal and the chiral amines (sec-butylamine, methylbenzylamine and amphetamine). The use of nuclear Overhauser effect (NOE) has allowed to observe that the ends of the molecules were close proving a quasi-folded conformation. The latter was further confirmed by theoretical calculations The authors have concluded that the conformational locking explains the good separation of 13C NMR signals between the diastereomeric imines obtained allowing a good determination of the enantiomeric excess using the open chain (S)-citronellal.

I find the paper very interesting in the area of structural chemistry especially using rare NMR techniques. One scientific advise that I really want to see in this article, it is the use of 2D-NOESY experiment performed on the compounds trying to look the correlation established between the imine proton and the ending methyl group of (S)-citronellal chain. By this way, the authors will clearly confirm their findings using 1D-NOE and will further decorate the experimental results.

One question, could the resulting imines crystallize well ? it i s then possible to make single-crystl X-ray diffraction to confirm the proposed conformation.  

Anyway, the article seems well presented for publication.

Author Response

At the suggestion of the reviewer, we performed 2D-NOESY experiment using the pulse sequence noesyph with the parameters TD = 2048 (F2) and 256 (F1), DS = 4 and NS = 24 (Figures 1, 2 and 3). However, 2D-NOESY presented some artifacts and noise that could make signal attribution inaccurate, which seems 1D-NOESY is a better choice, providing cleaner spectra with an easy and direct interpretation. As mentioned on the article of Butts and co-workers (Org. Biomol. Chem., 2011, 9, 177): “in summary, we find the 1D-NOESY experiment to be faster, simpler, at least as accurate as 2D-NOESY for internuclear distance determinations, and able to applied with more confidence to longer range NOE contacts.”

We know that the usual choice is 2D-NOESY with zero-quantum suppression, but exceptions would be if you have a very concentrated sample and you are only interested in one or two NOEs and the peaks to be irradiated are well-separated, as is our case.

The purpose of running a NOESY is often to determine conformation by establishing a distance between two protons. When more than one conformation is present, however, the NOE may give misleading distances. If the conformations are being averaged over the time scale of the experiment (the mixing time) the NOE will not reflect the average distance between the protons but rather the average of the inverse sixth power of the distance.

 Figure 4. 2D NOESY (pulse sequence noesyph, T = 273 K) from imines 5S,R and 5S,S.

Figure 5. 2D NOESY (pulse sequence noesyph, T = 273 K) from imines 6S,R and 6S,S.

Figure 6. 2D NOESY (pulse sequence noesyph, T = 273 K) from imines 7S,R and 7S,S.

In the mixture of the reactants (citronellal and amine), we do not observe the formation of crystals, even without the addition of any solvent. In this step, we see only a turbid solution that becomes 

Round 2

Reviewer 2 Report

I appreciated that the authors answered all my comments. I have only a minor point to address:

- the sentence "Therefore, we conclude that the NMR signal of diastereomeric derivatives (S,S and S, R imines) are well separated because the conformers do not differ significantly due to the quasi-folded conformation caused by London forces and interactions between molecular orbitals." reported in line 233-235 remains unclear to me. The NMR signals are normally well separated if conformers differ significantly, can  they better clarify this point?

- please make a check of the english:

i.e in the figure legend of NMR spectra, correct the sentence "Irradiated at 5.03 ppm (H11) and 0.73 ppm (H8)."

Author Response

As shown in Figure 2 of the article (shown below to facilitate understanding), the NMR signal is a sum of all NMR signals of the conformers of the same molecule, also taking into account the percentage of Boltzmann population of each conformer.

                                              Figure 2. Representation of NMR signals and chemical shift.

Thus, if the conformers are similar, the NMR signal becomes thinner, that is, more resolved, because all the colored lines in Figure 2 are closer together. If the conformers differ greatly, each conformer will have a relatively different chemical shift from each other, resulting in a wider final signal (resulting from all conformer), that is, less resolved.

Therefore, when we say that the signals are well separated we have to take into consideration not only the chemical shift absolute value, but also the resolution of these signals.

In our case, poor resolution interferes in the signal quantification, since the integrals of both signals are not well defined. See figure below.

Figure 3. representation of the difference in signals.

Regarding the correction of English, the article was submitted for review available by the publisher MDPI of Molecules journal and will be ready within two days.
